# Novel Monomethoxy Poly(Ethylene Glycol) Modified Hydroxylated Tung Oil for Drug Delivery

**DOI:** 10.3390/polym15030564

**Published:** 2023-01-21

**Authors:** Huafen Wang, Huanhuan He, Jiaxiang Zhang, Juntao Liu, Yuwei Zhuang, Yuanyuan Yin, Zhiyong Ren, Yang Fu, Suqin He

**Affiliations:** 1High & New Technology Research Center of Henan Academy of Sciences, No. 56 Hongzhuan Road, Zhengzhou 450002, China; 2School of Materials Science and Engineering, Zhengzhou University, Zhengzhou 450001, China; 3Faculty of Science, Henan University of Animal Husbandry and Economy, Zhengzhou 450001, China

**Keywords:** hydroxylated tung oil, poly(ethylene glycol), drug delivery

## Abstract

Novel monomethoxy poly(ethylene glycol) (*m*PEG) modified hydroxylated tung oil (HTO), denoted as *m*PEG-HTO-*m*PEG, was designed and synthesized for drug delivery. *m*PEG-HTO-*m*PEG consists of a hydroxylated tung oil center joined by two *m*PEG blocks via a urethane linkage. The properties of *m*PEG-HTO-*m*PEG were affected by the length of the mPEG chain. Three *m*PEG with different molecular weights were used to prepare *m*PEG-HTO-*m*PEG. The obtained three *m*PEG-HTO-*m*PEG polymers were characterized by nuclear magnetic resonance (NMR), Fourier transformation infrared spectroscopy (FT-IR), differential scanning calorimetry (DSC) and gel permeation chromatography (GPC), respectively. Furthermore, the particle sizes of *m*PEG-HTO-*m*PEG micelles were evaluated by dynamic light scattering (DLS) and transmission electron microscope (TEM). A critical aggregation concentration (CAC) ranged from 7.28 to 11.73 mg/L depending on the chain length of mPEG. The drug loading and release behaviors of *m*PEG-HTO-*m*PEG were investigated using prednisone acetate as a model drug, and results indicated that hydrophobic prednisone acetate could be effectively loaded into *m*PEG-HTO-*m*PEG micelles and exhibited a long-term sustained release. Moreover, compared with HTO, *m*PEG-HTO-*m*PEG had no obvious cytotoxicity to HeLa and L929 cells. Therefore, monomethoxy poly(ethylene glycol) modified hydroxylated tung oil *m*PEG-HTO-*m*PEG may be a promising drug carrier.

## 1. Introduction

As an important and promising drug delivery system, self-assembled polymer micelles based on amphiphilic polymers have received increasing attention recently [1,2,3]. The amphiphilic polymers can easily self-assemble into core-shell structures in aqueous solutions [4]. The hydrophobic polymeric core can improve the solubility of the insoluble drugs and prolong the drug release, thereby reducing side effects and improving the therapeutic effect [5].

Vegetable oils, extracted primarily from the seeds of oil-seed plants, are considered to be the most potent resource for green polyols due to their low price, abundant availability, inherent biodegradability and many chemical sites with potential reactivity [6,7,8,9,10]. Specifically, tung oil, also known as China wood oil, is an environmentally friendly, biodegradable, renewable natural drying oil [11,12]. In general, it is extracted from the seeds of the tung tree nut, and the main component is a triglyceride based on alpha-eleostearic acid (77–84%) with three conjugated carbon-carbon double bonds (9–10 cis, 11–12 trans, 13–14 trans), while the other components include oleic acid (3.5–12.7%) with one double bond and linoleic acid (8–10%) with two non-conjugated carbon-carbon double bonds [13,14,15,16]. The long alkyl chains of tung oil are suitable as hydrophobic segments of amphiphilic polymers [17,18].

PEG (poly(ethylene glycol)) has become a crucial ingredient in the modification and preparation of amphiphilic polymers for drug delivery due to its advantage in escaping reticuloendothelial system (RES) capture. The benefit of this is that drug-loaded micelles circulate long in blood vessels during transportation [19,20]. Its non-toxicity, non-immunogenic and good biocompatibility have also attracted much attention. Furthermore, PEG is FDA-approved for use in intravenous, oral, and dermal applications in humans [21,22,23].

The aim of this work was to prepare novel monomethoxy poly(ethylene glycol) modified hydroxylated tung oil, denoted as *m*PEG-HTO-*m*PEG, and to investigate its drug loading and release behavior. *m*PEG-HTO-*m*PEG was synthesized from hydroxylated tung oil and monomethoxy poly(ethylene glycol) using hexamethylene diisocyanate (HDI) as a coupling agent (Figure 1). The chemical structures, molecular weights and thermal properties of the obtained products were fully characterized by nuclear magnetic resonance (NMR), Fourier transformation infrared spectroscopy (FT-IR), differential scanning calorimetry (DSC) and gel permeation chromatography (GPC). The spontaneous self-assembly and micellization behavior of *m*PEG-HTO-*m*PEG in an aqueous solution was evaluated by fluorescence probe techniques, dynamic light scattering (DLS) and transmission electron microscope (TEM). Moreover, prednisone acetate was investigated as a modal drug to investigate drug loading and release. We expect that the designed novel monomethoxy poly(ethylene glycol) modified hydroxylated tung oil will be a promising drug carrier in the biomedical field in the future.

## 2. Materials and Methods

### 2.1. Materials

Hydroxylated Tung Oil (HTO) was generously provided by US A-line Company. Hexamethylene Diisocyanate (HDI) was purchased from Wanhua Chemical Group Co., Ltd., Yantai, Shandong, monomethoxy Poly(ethylene glycol) (*m*PEG, *M*_n_ = 550, 750 and 1000; Shanghai Aladdin Biochemical Technology Co., Ltd., Shanghai, China) was vacuum dried at room temperature prior to use. Before use, toluene was distilled under Na/K alloys from Shanghai Chemical Reagent Company (Shanghai, China). Dibutyltindilaurate (DBTDL, a catalyst) was purchased from Shanghai Macklin Biochemical Co., Ltd. (Shanghai, China). The prednisone acetate was purified from prednisone acetate tablets (Zhejiang Xianju Pharmaceutical Co., Ltd., Taizhou, China). In order to obtain prednisone acetate, the prednisone acetate tablets were ground and dissolved in chloroform, the starch was removed by filtration, and the solvent was removed. Tetrahydrofuran (THF), dimethyformamide (DMF), dimethyl sulfoxide (DMSO) were supplied by Fu Yu Fine Chemical Co., Ltd. (Tianjin, China), The human cervix carcinoma (HeLa), fibroblast (L929) were originally obtained from American Type Culture Colection (Shanghai Baiye Biotechnology Center, Shanghai, China) and then incubated under 37 °C within 5% CO_2_ atmosphere. Dulbecco’s modified eagle medium (DMEM), fetal bovine serum (FBS) and dulbecco’s phosphate buffered saline (PBS) were purchased from Giboco. 3-(4,5-dimethy1-2-thiazoyl)-2,5-diphenyl tetrazolium bromide (MTT) was purchased from Fluka (Buchs, Switzerland). The rest of the solvents and reagents were used without any further treatment as received.

### 2.2. Preparation of mPEG-HTO-mPEG

First, isocyanate-terminated HTO (NCO-HTO-NCO) was synthesized by the reaction of HTO with HDI in a molar ratio of 1:2.52 in toluene at 60 °C for 3 h. Specifically, 3.98 g HTO (0.01 mol) was dissolved in 20 mL anhydrous toluene in a 150 mL three-neck flask. Then a solution of 4.24 g HDI (0.0252 mol) in 20 mL anhydrous toluene was added dropwise into the flask under a dry nitrogen atmosphere. The reaction was carried out at 60 °C for 3 h. Finally, toluene was removed by rotary evaporation. The resulting NCO-HTO-NCO was degassed under vacuum for 3 h with stirring and used for the next reaction.

NCO-HTO-NCO was dissolved into toluene in a 150 mL three-neck flask and two drops of DBTDL were added as catalyst. Next, a solution of 0.02 mol *m*PEG (*M*_n_ = 550, 750 or 1000) dissolved in 20 mL anhydrous toluene was dropped into the reaction system. The reaction temperature was heated to 80 °C. The reaction was stopped when the isocyano-content approached 0. After removing the solvent and drying under vacuum, *m*PEG-HTO-*m*PEG was mainly obtained. During drug loading, by-product *m*PEG-HDI-*m*PEG and excess *m*PEG could be removed by dialysis method.

### 2.3. FT-IR Characterization

Fourier transform infrared (FT-IR) spectra of the obtained *m*PEG-modified hydroxylated tung oils were determined on a Nicolet 8700 spectrometer (Thermo Scientific, Waltham, MA, USA) between 4000 cm^−1^ and 500 cm^−1^ with the resolution of 4 cm^−1^. The samples were dissolved in dichloromethane (CHCl_2_) and cast on KBr plates for FT-IR characterization.

### 2.4. ^1^H NMR Characterization

^1^H NMR spectra of the obtained *m*PEG-modified hydroxylated tung oils were determined on an Agilent 400 NMR nuclear magnetic spectrometer (Agilent, Palo Alto, CA, USA) at a frequency of 400 Hz. The samples were dissolved in deuterated chloroform (CDCl_3_) or deuteroxide (D_2_O), and spectra were recorded at room temperature with tetramethyl silane (TMS) as an internal standard.

### 2.5. GPC Measurement

The molecular weights and molecular weight distributions of the obtained *m*PEG-modified hydroxylated tung oils were determined by Waters ALLIANCE e2695 system combined with a Waters-2414 refractive index detector, (Waters Corp., Milford, MA, USA). Poly(ethylene glycol) standards were used as calibration standards. The mobile phase was tetrahydrofuran (THF) flowing at 1.0 mL/min with 0.3% (*w*/*v*) sample concentration and 20 µL injection volume.

### 2.6. DSC Measurements

The DSC204 thermal analyzer (NETZSCH-Gerätebau GmbH, Selb, Germany) was used to conduct differential scanning calorimetry (DSC) measurements. All Samples were heated from room temperature to 100 °C, cooled form 100 to −70 °C and reheated at this temperature. Melting points were then recorded for the second heating cycle.

### 2.7. Preparation of mPEG-HTO-mPEG Micelles

The *m*PEG-HTO-*m*PEG micelles were prepared via membrane dialysis. Typically, *m*PEG-HTO-*m*PEG was completely dissolved in THF at an initial concentration of 200 mg/mL. After the resulting solution was transferred to a dialysis tube (MWCO: 1000 g/mol), it was dialyzed against 2 L deionized water for 24 h with vigorous stirring to remove organic solvents at room temperature. The deionized water was replaced every 4 h. As the dialysis solution changed from transparent to translucent, it indicated that *m*PEG-HTO-*m*PEG micelles were formed.

### 2.8. Determination of Critical Aggregation Concentration

Fluorescence spectra were carried out on a QW40 fluorescence spectrophotometer (Photon Technology International, Arizona, USA). As the hydrophobic fluorescence probe, pyrene was added to individual containers at concentrations of 10^−4^ mol/L in acetone (18 µL). After acetone was evaporated, three milliliters of *m*PEG-HTO-*m*PEG aqueous solutions with varying concentrations were added to the containers. The final concentration of pyrene was 6 × 10^−7^ M. In order to achieve equilibrium solubilization of pyrene in aqueous solutions, the solutions were maintained at room temperature for 24 h. At a wavelength of 393 nm, the excitation spectra were scanned from 300 to 360 nm. Both the excitation and emission bandwidths were 5 nm. The intensity ratio of the peak at 340 nm to that at 337 nm from excitation spectra was plotted against the logarithm of the *m*PEG-HTO-*m*PEG concentration. According to the plot, the critical aggregation concentrations (CACs) were calculated at low polymer concentrations.

### 2.9. Transmission Electron Microscopy (TEM) Observation

An 80 kV accelerated transmission electron microscope (TEM) (Hitachi HT7700, Tokyo, Japan) was used to investigate the micelle morphology and size. Before examination, micelle solution was deposited directly onto a copper grid with Formvar film and dried in air.

### 2.10. Size Distribution Measurement

The mean particle size and size distribution of self-assembled *m*PEG-HTO-*m*PEG micelles in the aqueous phase were determined using DLS by Zetasizer Nano ZEN3690 (Malvern Instruments Ltd., Marvin, UK). All measurements were performed at 25 °C. In order to measure the *m*PEG-HTO-*m*PEG aqueous solutions, they were passed through a 0.45 m pore-sized syringe filter.

### 2.11. Drug Loading

Prednisone acetate as a model drug was loaded into *m*PEG-HTO-*m*PEG micelles using dialysis method. The drug-loaded *m*PEG-HTO-*m*PEG micelles were prepared as follows: *m*PEG-HTO-*m*PEG (10 mg) and prednisone acetate (4 mg) were dissolved in the mixed solvent containing 9 mL THF and 1 mL *N,N*-Dimethylformamide (DMF) at room temperature. The solutions were dialyzed against 2 L deionized water for 24 h (MWCO: 1000 g/mol) and the deionized water was replaced every 4 h to remove organic solvents and unloaded drug.

### 2.12. In Vitro Drug Release Study

In vitro drug release test was conducted by immersing the dialysis tube in 10 mL of PBS (I = 0.1 M) at a pH of 7.4 at 37.4 °C after drug loading. The outer solution was withdrawn in aliquots of 3 mL at predetermined intervals, and fresh solution was added to maintain a constant volume. Using an experimentally obtained standard calibration curve, the amount of prednisone acetate released from the micelles at 37.4 °C was determined. Combined with the unreleased drug retained in the micelles, the total mass of drug loaded into the micelles contained the cumulative mass of drug released. After drug release, the micelles solution was lyophilized, then dissolved in 20 mL of DMF, followed by UV absorbance measurement at 270 nm to determine the amount of unreleased drug. In order to calculate the drug release efficiency (DRE), entrapment efficiency (EE) and drug loading efficiency (DLE), we formulated the formula as follows:DRE (wt.%) = (mass of drug released/mass of drug loaded in micelles) × 100%
EE (wt.%) = (mass of drug loaded in micelles/mass of drug fed initially) × 100% 
DLE (wt.%) = (mass of drug loaded in micelles/mass of drug loaded micelles) × 100% (1)

### 2.13. In Vitro Cytotoxicity Test

MTT assay was performed for in vitro cytotoxicity evaluation of *m*PEG-HTO-*m*PEG [24]. Briefly, a 96-well plate was seeded with 6 × 10^3^ HeLa cells or L929 cells in each well. Cells were incubated for 24 h (37 °C, 5% CO_2_) in medium containing *m*PEG-HTO-*m*PEG at various concentrations for 48 h. After this, fresh Dulbecco’s modified Eagle’s medium (DMEM) and 5 mg/mL MTT solution (3-(4,5-dimethylthiazol-2-yl)-2,5-diphenyl tetrazo-lium bromide) were added to the medium. The medium was removed after incubation for 4 h, and 200 μL of dimethyl sulfoxide (DMSO) was added to dissolve the formazan crystals. Cell viability was determined by measuring with a Microplate Reader Model 550 (Bio-Rad, Hercules, CA, USA) at a wavelength of 570 nm. The cell viability was calculated from:Cell Viability (%) = (OD_treated_/OD_control_) × 100

## 3. Results and Discussion

### 3.1. Synthesis and Characterization of mPEG-HTO-mPEG

The *m*PEG modified hydroxylated tung oils were synthesized via the coupling reaction between hydroxyl groups of *m*PEG with various *M*_n_ and isocyanate groups in NCO-HTO-NCO, wherein NCO-HTO-NCO was prepared by the coupling reaction of HTO with excess HDI, as illustrated in Figure 1. The chemical structures and molecular weights of the obtained *m*PEG modified hydroxylated tung oils were characterized by FT-IR, ^1^H NMR and GPC.

The FT-IR spectra of NCO-HTO-NCO and *m*PEG-HTO-*m*PEG were shown in Figure 1. The typical absorption peak at 2273 cm^−1^ was assigned to -NCO groups in Figure 1A. The absorption band at 1719 cm^−1^ was ascribed to the carbonyl of urethane and there were no hydroxyl groups. These results indicated that HDI completely reacted with hydroxyl groups in HTO to form NCO-HTO-NCO. Meanwhile, the 2273 cm^−1^ peak was disappeared in Figure 1B–D. Around 3340 cm^−1^, a broad stretching band appeared mainly due to hydrogen-bonded N-H stretching vibrations. As a result of vibrational coupling with N-H deformation vibration and C-N stretching vibration, the peaks observed at 1532 cm^−1^ and 1248 cm^−1^ belonged to Amid II and Amid III bands, respectively [25]. It was believed that the absorption peaks at 2923 cm^−1^ and 2863 cm^−1^ were caused by the asymmetric and symmetric stretching vibrations of methylene. In *m*PEG, the absorption peaks at 1105 cm^−1^ were attributed to ether group (C-O-C) asymmetric stretching vibrations. The peak at 950 cm^−1^ represents characteristic peak of double bond in HTO. These results indicated that *m*PEG has been successfully introduced into the *m*PEG-HTO-*m*PEG chains.

The ^1^H NMR spectra of NCO-HTO-NCO and a representative *m*PEG-HTO-*m*PEG-2 were shown in Figure 2. As shown in Figure 2A, the HDI produced four different methylene proton signals: 3.13 ppm (4H, m, H-m, -CH_2_-CH_2_-CH_2_-NH-COO-), 1.47 ppm (8H, m, H-n, -CH_2_-CH_2_-CH_2_-NH-COO-), 1.24 ppm (8H, m, H-c, -CH_2_-CH_2_-CH_2_-NH-COO-), 3.31 ppm (4H, m, H-o, -CH_2_-CH_2_-CH_2_-CH_2_-NCO). The peak at 5.10 ppm belonged to H-l (2H, t, -NH-COO-). These results indicated that NCO-HTO-NCO was successfully synthesized.

In Figure 2B, the peaks at 3.36 ppm and 3.62 ppm attributed to the methyl protons at the end of the *m*PEG chains (6H, s, CH_3_-O) and the oxyethylene unit protons in *m*PEG (br, m, -CH_2_CH_2_O-), respectively. The peak at 0.86 ppm was ascribed to the methyl group at the end of *m*PEG-HTO-*m*PEG-2.The peak at 2.29 ppm belonged to α-H that was attached to a carbonyl group (-CH_2_-CH_2_CO) and another peak at 1.58 ppm belonged to β-H (-CH_2_-CH_2_CO). The peak at 4.89 ppm was attributed to a methine proton(H-j). The peak at 4.0 ppm should belong to the methylene protons (H-q). The peak at 4.30 ppm was attributed to methylene group connected with a methine proton whereas the peaks at 5.33–6.38 ppm were in the chemical shift range of allyl hydrogen. These results confirmed that *m*PEG-HTO-*m*PEG was successfully synthesized. This was in good agreement with the FTIR results.

The molecular weights and molecular weight distributions of the obtained three *m*PEG-HTO-*m*PEG polymers were measured by GPC using THF as the mobile phase (Table 1). The calculated number-based average molecular weights of *m*PEG-HTO-*m*PEG-1, *m*PEG-HTO-*m*PEG-2, and *m*PEG-HTO-*m*PEG-3 are in good accordance with the theoretical molecular weight and were found to be 2111 g mol^−1^, 2435 g mol^−1^, and 2200 g mol^−1^, respectively. Taken together, above FT-IR, ^1^H-NMR and GPC results indicated that the synthesis *m*PEG-HTO-*m*PEG was successful. The GPC traces of the polymers were shown in Appendix A.

### 3.2. Thermal Properties of mPEG-HTO-mPEG

To investigate the thermal properties of *m*PEG-HTO-*m*PEG, DSC measurements were employed. All the *m*PEG-modified hydroxylated tung oils were heated from room temperature to 100 °C, cooled from 100 to −70 °C and reheated at this temperature, in order to produce maximum crystallinity. Figure 3 showed the obtained DSC heating scans and melting points from the second heating cycle. A melting peak appears during the heating process when the molecular weight of the *m*PEG is less than 1000. When compared with curve 3A’–B’, curves 3A and 3B have one more peak because there is a phase separation between soft PEG segments and hard segments containing HTO and HDI segments [26]. A single melting endotherm at around 37 °C was detected for *m*PEG-HTO-*m*PEG-3, which belongs to the melting transition temperature (T_m_) of *m*PEG1000 segments, which indicates that no obvious phase separation occurs.

### 3.3. Self-Assembly of mPEG-HTO-mPEG Micelles

Aqueous micelles with a core-shell structure are formed by the amphiphilic nature of *m*PEG-HTO-*m*PEG, which consists of two hydrophilic *m*PEG segments and one hydrophobic HTO segment. The CAC was considered an important parameter describing the micelle’s stability. The CACs of the obtained three *m*PEG-modified hydroxylated tung oils was estimated by fluorescence spectra using pyrene as a hydrophobic probe. It was found the absorption band had a red-shift from 337 to 340 nm with increasing the concentration of *m*PEG-HTO-*m*PEG, indicating that pyrenes were transferred into the hydrophobic core and the formation of micelles [27,28]. The CACs were calculated by plotting the intensity ratio of I_340_/I_337_ from the excitation spectra of pyrene against the log of *m*PEG-HTO-*m*PEG concentrations (Figure 4). The CACs were obtained from the intersection of the two tangent lines. As shown in Figure 4, the CAC values for *m*PEG-HTO-*m*PEG-1, 2, 3 were 7.28 mg/L, 8.75 mg/L and 11.73 mg/L, respectively. Under diluted conditions, *m*PEG-HTO-*m*PEG retained a nanomicellar structure, indicating that the polymeric micelles formed were thermodynamically stable and may be used as drug carriers. Moreover, the *M*_n_ or chain lengths of *m*PEG had a distinct effect on the CAC values. The higher the *M*_n_ of *m*PEG or the longer the length of *m*PEG segments, the higher the CAC values. A major contributing factor was the extended hydrophilic chains in *m*PEG, which enhanced hydrophilicity.

The sizes, size distributions and morphologies of *m*PEG-HTO-*m*PEG micelles were measured by DLS and TEM as shown in Figure 5. It can be seen from Figure 5A that the hydrodynamic diameter of *m*PEG-HTO-*m*PEG-2 micelles determined by DLS was 157 nm, and the size distribution index was 0.228. As can be seen in Figure 5B, the micelles were spherical and well dispersed as individual particles with a diameter of around 56 nm. Obviously, the particle size measured by TEM was generally smaller than that measured by DLS. That was because the former showed the morphology of micelles when dehydrated, while the latter measures their hydrodynamic diameter in water. Both DLS and TEM results demonstrated that *m*PEG-HTO-*m*PEG could self-assemble into polymeric micelles in aqueous solution. Notably, the size and the spherical structure of micelles were suitable for drug delivery.

Figure 2C shows the ^1^H NMR spectrum of *m*PEG-HTO-*m*PEG-2 in D_2_O, which provided an insight into the micellar form. As shown, the HTO and HDI segments were almost lost with the spectrum in CDCl_3_ (Figure 2C). The cause of this phenomenon was the amphiphilic polymers could easily undergo phase separation in the different selected solvents. Therefore, *m*PEG segment could be well solvated in D_2_O and detectable by ^1^H NMR. The hydrophobic segments were lost in the micelle core. This phenomenon has been found in our previous literature [24]. Thus ^1^H NMR analysis of *m*PEG-HTO-*m*PEG-2 indicated the samples could be self-assembled into micellar in which hydrophobic core consisted of HTO and HDI segments and the hydrophilic shell was *m*PEG segment. In addition to TEM observations and DLS analyses, NMR results showed that *m*PEG-HTO-*m*PEG self-assembled into nanosized micelles in an aqueous medium.

### 3.4. Drug Loading and In Vitro Drug Release

To evaluate the potential of *m*PEG-HTO-*m*PEG nanomicelles as a vehicle for the delivery of hydrophobic drugs. Prednisone acetate was used as a model drug. Prednisone acetate has been used for treating certain inflammatory diseases and cancers. It is a corticosteroid drug that is particularly effective as an immunosuppressant [24,29]. However, prednisone acetate suffers from its poor solubility and significant side effects. To address these issues, prednisone acetate was loaded into *m*PEG-HTO-*m*PEG micelles using a dialysis method. Table 2 summarized the entrapment efficiency (EE) and drug loading efficiency (DLE) of prednisone acetate-loaded *m*PEG-HTO-*m*PEG micelles. According to Table 2, prednisone acetate loading efficiency declines with increasing *M*_n_ or length of hydrophilic *m*PEG, which corresponds to a decrease in the length [30]. This phenomenon has also been found in previous literature [31]. The low drug entrapment efficiency in this work was mainly caused by the excess prednisone acetate in the initial inventory, which was waste due to the polymeric micelles cannot hold too much drug. So prednisone acetate should be added in the appropriate amount to the polymeric micelles to make the drug-loaded micelles [32].

A polymeric micelle loaded with prednisone acetate was fabricated and examined in PBS (0.1 M, pH = 7.4) at 37.4 °C in vitro. As shown in Figure 6, it indicated a sustained release of prednisone acetate in PBS (0.1 M, pH = 7.4). The release rate was relatively fast in the initial stage of drug release and slowed down over a prolonged time. The final release percentage of prednisone acetate was 25.1%, 20.3% and 17.5% for *m*PEG-HTO-*m*PEG-1, 2, 3 micelles, respectively. It was noticed that the *m*PEG-HTO-*m*PEG micelles with higher *m*PEG contents or *M*_n_ had slower drug release rate, likely due to the cross-linking in the core of micelles. The cross-linking may occur in the core because of the three conjugated carbon-carbon double bonds in the hydrophobic HTO segment [18,33]. This could result in drug release decrease. Another reason for slower release might be crystallinity of prednisone acetate. Previous studies reported that prednisone acetate is released slowly from micelles due to their hydrophobic and highly crystalline natures [32]. Additionally, the interaction between prednisone acetate and the core-forming HTO segment may be a significant factor in determining the release rate of the drug. [34].

### 3.5. In Vitro Evaluation of Cell Viability

The in vitro cytotoxicity of the *m*PEG-HTO-*m*PEG polymers against HeLa cells and L929 cells was evaluated by MTT assay. As shown in Figure 7, HTO showed high cytotoxicity (30–40% cell viability) at concentrations of up to 100 mg/L. However, the cell viabilities of L929 cells incubated with three *m*PEG-HTO-*m*PEG polymers were about 80% at concentrations up to 100 mg/L, and the cell viabilities of HeLa cells were over 80% at all test concentrations up to 140 mg/L, indicating the low cytotoxicity and good biocompatibility of the *m*PEG modified hydroxylated tung oils.

## 4. Conclusions

In summary, the prednisone acetate-loaded micelles based on novel monomethoxy poly(ethylene glycol) modified hydroxylated tung oils were successfully prepared by self-assembly in aqueous solution. DLS and TEM results showed that novel monomethoxy poly(ethylene glycol) modified hydroxylated tung oils could self-assemble to form micelles with a well-defined spherical shape ranging from 40 to 70 nm in diameter. The CAC values and drug release rate decreased with increasing *M*_n_ of *m*PEG chains. Cell cytotoxicity tests showed that *m*PEG-HTO-*m*PEG polymers had no obvious cytotoxicity to HeLa and L929 cells. Therefore, the novel monomethoxy poly(ethylene glycol) modified hydroxylated tung oils may have great potential as drug carriers.

## 5. Patents

This research supported Chinese patent ZL201810178431.5 (Title: The preparation and drug-loaded micelles of Glyceryl monostearate-poly(ethylene glycol)_2_.

## Data Availability

Not applicable.

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
