# Peer review of "Novel Monomethoxy Poly(Ethylene Glycol) Modified Hydroxylated Tung Oil for Drug Delivery"

_polymers, 2023, doi:10.3390/polym15030564_

Round 1
Reviewer 1 Report
The Author should revise the following suggested points
1. In the discussion part of the manuscript author should compare the drug loading efficiency of previously reported similar systems.
2. figure no 3 the DSC thermogram why curve ( endotherm ) while it upward direction generally it should be the opposite of it with an endotherm.
3. Author should include the images of Cells for In vitro evaluation of cell viability
4. Author should cite the following very relevant previous research articles in the introduction and discussion. A. Effect of polyethylene glycol on properties and drug encapsulation–release performance of biodegradable/cytocompatible agarose–polyethylene glycol–polycaprolactone amphiphilic co-network gels." ACS applied materials & interfaces 8.5 (2016): 3182-3192. B.. Synthesis and multi‐responsive self‐assembly of cationic poly (caprolactone)–poly (ethylene glycol) multiblock copolymers. Chemistry–A European Journal, 23(34), 8166-8170. C.Self-assembly of partially alkylated dextran-graft-poly [(2-dimethylamino) ethyl methacrylate] copolymer facilitating hydrophobic/hydrophilic drug delivery and improving conetwork hydrogel properties." Biomacromolecules 19.4 (2018): 1142-1153.
5. Author should include the GPC traces of the copolymers in the manuscript.
6.
Author Response
Dear editor and reviewer:
Thank you very much for your valuable comments. We have revised and added relevant contents carefully according to the requirements of the reviewer. See the revision manuscript and an attachment for details.

Reviewer 2 Report
Attached.

Author Response

(The authors gave the same response as above.)
